# Antibacterial and Antiviral Effects of Ag, Cu and Zn Metals, Respective Nanoparticles and Filter Materials Thereof against Coronavirus SARS-CoV-2 and Influenza A Virus

**DOI:** 10.3390/pharmaceutics14122549

**Published:** 2022-11-22

**Authors:** Anna-Liisa Kubo, Kai Rausalu, Natalja Savest, Eva Žusinaite, Grigory Vasiliev, Mihkel Viirsalu, Tiia Plamus, Andres Krumme, Andres Merits, Olesja Bondarenko

**Affiliations:** 1Laboratory of Environmental Toxicology, National Institute of Chemical Physics and Biophysics, Akadeemia tee 23, 12618 Tallinn, Estonia; 2Nanordica Medical OÜ, Vana-Lõuna 39a-7, 10134 Tallinn, Estonia; 3Institute of Technology, University of Tartu, Nooruse 1, 50411 Tartu, Estonia; 4Laboratory of Polymers and Textile Technology, Department of Materials and Environmental Technology, Tallinn University of Technology, Ehitajate tee 5, 19086 Tallinn, Estonia

**Keywords:** coronavirus, silver, zinc oxide, copper oxide, toxicity, face masks, electrospinning

## Abstract

Due to the high prevalence of infectious diseases and their concurrent outbreaks, there is a high interest in developing novel materials with antimicrobial properties. Antibacterial and antiviral properties of a range of metal-based nanoparticles (NPs) are a promising means to fight airborne diseases caused by viruses and bacteria. The aim of this study was to test antimicrobial metals and metal-based nanoparticles efficacy against three viruses, namely influenza A virus (H1N1; A/WSN/1933) and coronaviruses TGEV and SARS-CoV-2; and two bacteria, *Escherichia coli* and *Staphylococcus aureus*. The efficacy of ZnO, CuO, and Ag NPs and their respective metal salts, i.e., ZnSO_4_, CuSO_4_, and AgNO_3_, was evaluated in suspensions, and the compounds with the highest antiviral efficacy were chosen for incorporation into fibers of cellulose acetate (CA), using electrospinning to produce filter materials for face masks. Among the tested compounds, CuSO_4_ demonstrated the highest efficacy against influenza A virus and SARS-CoV-2 (1 h IC50 1.395 mg/L and 0.45 mg/L, respectively), followed by Zn salt and Ag salt. Therefore, Cu compounds were selected for incorporation into CA fibers to produce antiviral and antibacterial filter materials for face masks. CA fibers comprising CuSO_4_ decreased SARS-CoV-2 titer by 0.38 logarithms and influenza A virus titer by 1.08 logarithms after 5 min of contact; after 1 h of contact, SARS-COV-2 virus was completely inactivated. Developed CuO- and CuSO_4_-based filter materials also efficiently inactivated the bacteria *Escherichia coli* and *Staphylococcus aureus*. The metal NPs and respective metal salts were potent antibacterial and antiviral compounds that were successfully incorporated into the filter materials of face masks. New antibacterial and antiviral materials developed and characterized in this study are crucial in the context of the ongoing SARS-CoV-2 pandemic and beyond.

## 1. Introduction

The recent emergence of severe acute respiratory syndrome coronavirus 2 (SARS-CoV-2), the causative agent of COVID-19, has tremendous worldwide socioeconomic impacts. The genome of SARS-CoV-2 shares about 80% identity with that of SARS-CoV-1, a coronavirus causing SARS disease. SARS-CoV-2 is a positive-strand enveloped RNA virus, with the size of the virion ca. 125 nm in diameter. The spike glycoprotein S on the surface of the virion determines the attachment and the entry of the virus to the host cells [1,2,3]. The binding affinity of the SARS-CoV-2 S protein to its cellular receptor hACE2 is higher as compared to that of S protein of the SARS-CoV-1 [4], making SARS-CoV-2 extremely transmittable [5,6,7,8]. To prevent virus transmission, it is suggested by the World Health Organization (WHO) to wear masks for avoiding the nosocomial transmission of the virus. Moreover, according to the WHO and the Centers for Disease Control and Prevention, wearing of masks is a relevant measure to reduce the spread of the coronavirus [9,10,11].

SARS-CoV-2 is an enveloped virus, similar to the Zika virus (ZIKV), Ebola virus (EBOV), SARS-CoV-1, and MERS-CoV [12], bearing a lipid envelope derived from the host cell membranes. Traditionally, antiviral agents and disinfectants are used to reduce the spread and inactivate these viruses [13]. To tackle the viral pandemics similar to COVID-19 through the implementation of materials comprising metals is a promising means [14,15,16,17,18]. Metals such as silver, copper, and zinc are well-known antimicrobials against a wide array of bacteria [19,20] and viruses, including human influenza A virus and SARS-CoV-2 [21,22,23,24,25,26,27,28]. The use of nanoparticles (NPs; size range, 1–100 nm) instead of metal salts enables control over metal ion release, prevents the rapid inactivation of metal ions, and enhances the antimicrobial effect of metal ions [19,29]. In particular, metal-based NPs are well-known for their antibacterial and antiviral properties, showing a significant response even at low concentrations, due to their high surface-to-volume ratio [19,30,31,32]. In the recent article by Bahrami et al., the authors reviewed the effects of a wide range of metal-based NPs on SARS-CoV-2 and showed that Ag NPs inhibited virus entry, reduced oxidative stress, and inhibited viral RNA synthesis. Cu NPs, on the other hand, damaged the virus membrane, destroyed the RNA, and interfered with functional proteins [21].

The outstanding virucidal efficacy of metal-based NPs makes them excellent materials to be applied on common surfaces toward impairing infection spread. Antimicrobial surface-coating materials comprising copper, silver, and zinc have shown good virucidal efficacy in controlling the transmission of viruses such as influenza A virus, human immunodeficiency virus type 1 (HIV-1), dengue virus type 2 (DENV-2), and human herpesvirus 1 (HHV-1) [14,33]. Considerable loss of human coronavirus HuCoV-229E infectivity was observed on copper-containing surfaces after 30 min of contact [34]. Surfaces treated with copper are already used in hospitals for the prevention of the spread of microbial contaminations and potentially viral transmissions [35].

Recently, many organic compounds of polymers, including biopolymers, have shown potential as antibacterial and antiviral agents to impair the spread of infectious diseases [36]. These polymer matrices are excellent as carriers of metal NPs and metals. Metals can be incorporated in the materials permanently, and these metal–polymer composites are efficient antimicrobials. Polymers and fiber materials thereof are used as components of fabrics and textiles and filter materials. Traditionally, the incorporation of the metal additives in the polymer matrices is achieved by the coating of the metals on the surface of the fibers or impregnation in the blend of polymer fibers [37]. Mostly, these materials are synthetic. For example, common surgical masks consist of plastic-based polymers such as polypropylene, polyurethane, polyacrylonitrile, polystyrene, polycarbonate, polyethylene, or polyester [38]. However, bio-based polymers are preferred for the future developments of the new biodegradable materials.

In this study, we tested the antibacterial and antiviral properties of the salts of Cu, Ag, and Zn; and respective NPs to develop new effective antimicrobial biomaterials thereof. To test the antibacterial properties of the compounds and materials, *E. coli* and *S. aureus* were selected, and for the antiviral properties, influenza A virus and coronaviruses transmissible gastroenteritis virus (TGEV) and SARS-CoV-2 were selected. The metal-based components with the best antimicrobial properties were incorporated into biopolymer CA, which is a natural environmentally friendly bio-based polymer. As the main results of the study, we provided valuable data on toxicity of studied metals and NPs to different bacteria and viruses and developed novel antimicrobial CuSO_4_-based filter material that inactivates SARS-CoV-2 within minutes of exposure.

## 2. Materials and Methods

### 2.1. Materials and Chemicals

CuO-NH_2_ and CuO-COOH were obtained from PlasmaChem GmbH (Germany). Ag NPs (nAg) were obtained from Sigma-Aldrich. CuO and CuSO_4_ were purchased as powders from Sigma-Aldrich and Alfa Aesar, respectively. AgNO_3_ was obtained from J.T.Baker, and ZnSO_4_ and ZnO from Sigma-Aldrich.

Cellulose acetate (CA) in granules with an average molecular mass of 30,000 g/mol was purchased from Aldrich Chemistry. Acetonitrile and dimethyl acetamide (DMA_c_) were from Sigma-Aldrich. Thymol (laboratory reagent grade), which was used as an additive to increase the porosity of electrospun filter materials, was obtained from Fisher Chemistry, and HNO_3_ was from Seastar Chemicals.

### 2.2. Viruses and Cells

The human influenza A virus, strain A/WSN/1933 (H1N1), was kindly provided by Prof. Denis Kaynov (Norwegian University of Science and Technology, NTNU, Trondheim, Norway). Transmissible gastroenteritis virus (TGEV) was a kind gift from Prof. Luis Enjuanes (Centro Nacional de Biotecnología-Consejo Superior de Investigaciones Científicas, CNB-CSIC, Madrid, Spain). SARS-CoV-2 Estonian isolate 3049 was propagated from a nasopharyngeal swab of RT-qPCR-positive patient’s sample. MDCK-2 (Madin-Darby canine kidney, the American Type Culture Collection code CCL-34) cells were used for propagation and titration of influenza virus; Vero-E6 (African green monkey, ((ATCC)) code CRL-1586) cells were used for propagation and titration of SARS-CoV-2, and ST cells (a kind gift from Prof. Luis Enjuanes’ laboratory) were used for propagation and titration of TGEV. All cells were grown in DMEM medium (Corning) containing 10% heat-inactivated fetal bovine serum (FBS, PAN Biotech), in the presence of 100 units/mL penicillin and 100 µg/mL streptomycin (Sigma).

Influenza A virus was grown in MDCK-2 cells. The virus growth medium (VGM) was DMEM, containing 0.2% bovine serum albumin (BSA, Sigma), 100 units/mL penicillin, 100 µg/mL streptomycin, and 1 µg/mL TPCK-treated (N-tosyl-L-phenylalanine chloromethyl ketone) trypsin (Sigma). TGEV was grown in ST cells in VGM, which was DMEM that contained 0.2% BSA, 100 units/mL penicillin, and 100 µg/mL streptomycin. SARS-CoV-2 was propagated in Vero-E6 cells in a VGM (DMEM complemented with 0.2% BSA and penicillin/streptomycin).

### 2.3. Dispersion Analysis for Polymer Solvent Systems with Metal Nanoparticles

CA was used as a polymer matrix for electrospinning solution preparation. Two solvents, acetonitrile and DMA_c_ were used to prepare solvent dispersions. NP and metal additives were used for CA polymer solutions to enhance the antibacterial and antiviral properties of the filter materials produced by electrospinning. CA polymer filter materials were electrospun in acetone–DMA_c_ solvent systems at 2:1 and 3:1 ratios. The solvent systems with metal compounds were characterized for dispersion by using dynamic light scattering (DLS) for evaluating the profile for size distribution for the particles and CA polymers in solution and provide additional information about the compatibility and quality of the NP polymer–solvent systems (Appendix A).

### 2.4. Electrospinning of Filter Materials

CA polymer solutions were prepared in an acetonitrile–DMA_c_ mixture in a mass mixing ratio of 2:1. The concentration of CA in the solvent mixture was 17 wt.%. To obtain a homogeneous solution, the added polymer was dissolved in the solvent mixture by mechanical mixing with the magnetic stirrer, at room temperature, for 24 h. The additives of CuO and CuSO_4_ in total mass concentration of 12.5 wt.% and 18.75 wt.% were added, achieving 7.5% and 10% Cu concentrations in polymer, respectively, and the solutions were stirred again for 24 h to achieve the homogeneity. To avoid precipitation of CuSO_4_ salt in CA solution, the ultrasonication treatment was applied for another 30 min using probe sonication (Branson Sonifier 450).

The electrospinning process was performed by using an in-house built-laboratory-scale horizontal electrospinning setup with cylindrical rotating collector covered with the non-woven fiber material with a 25 g/m^2^ density. The operating parameters for electrospinning are summarized in Appendix A. In brief, the electrospinning solution containing CA polymer in acetonitrile–DMA_c_ mixture in mass mixing ratio 2:1 was placed in a 2 mL syringe with a needle diameter of 0.4–0.6 mm. The solution was electrospun at 10 kV, with the distance of 15 cm between the collector and the needle. The pumping rate was from 0.4 up to 0.9 mL/h depending on the solution viscosity. The electrospinning was performed at room temperature and at an air humidity of 60%. The humidity of the isolated electrospinning box was controlled by using a portable humidity device, Air-O-Swiss.

#### 2.4.1. Analysis of Morphology of Electrospun Filter Material

The morphology of the filter materials was determined by high-resolution scanning electron microscope (SEM) Zeiss Ultra 55. For high-resolution imaging, in-lens secondary electron detection at an accelerating voltage (AV) of 4 kV was used. To study the chemical composition of samples, the back-scattered electron detector was used at 15 kV AV. Scanning transmission electron microscope (STEM) images were acquired by an FEI Titan Themis 200 microscope equipped with the Super X detector system for an energy-dispersive X-ray spectroscopy (EDX). For STEM measurements, biopolymer material was cut with scissors to fit in a 3 mm–diameter sample holder. Fibers were fixed by placing them between two Au grids.

#### 2.4.2. Air Permeability Testing of Electrospun Filter Materials

Air permeability testing of the materials was conducted according to the Medical Masks standard EVS-EN 14683:2019 (Estonian Centre for Standardisation and Accreditation). Before measuring, the filter mats were conditioned according to the Standard (Estonian Centre for Standardisation and Accreditation). The air permeability measurements were performed by using the measuring device FX 3340 MinAir (Textest Instruments). All measurements were performed on the sample area of 5.0 cm^2^ with 0.272 m/s air speed. The results are presented as the arithmetic average of 5 measurements with the standard deviation.

#### 2.4.3. Testing of Aerosol Filtration Efficiency of Electrospun Filter Materials

Aerosol filtration efficiency was measured according to the standards [39], using the aerosol generator TSI model 3076, the aerosol analyzer Scanning Mobility Particle Sizer (SMPS) TSI model 3082 + Condensation Particle Counter (CPC) TSI model 3775, the dryer TSI filtered air supply 3074B, silica gel dryer TSI 3062, and the deferential monometer CHY 886U. Polydisperse aerosol with a particle range of 11.8–429.4 nm was used for testing, covering a particle size of 300 nm, which N95 masks are certified to filtrate for [39].

#### 2.4.4. Hydrophobicity/Hydrophilicity Determining

The hydrophobicity/hydrophilicity of the filter materials was estimated by measuring the contact angle, using the Sessile drop method [40] with the device DataPhysics OCA 20 and SCA 20 software (DataPhysics Instruments GmbH, Filderstadt,·Germany). Distilled water was used as a liquid agent to create the drop on the measured surface of the material. All measurements were performed at room temperature and the air humidity at 40%.

#### 2.4.5. Quantification of Metal Content and the Released Metal Content of the Fiber Materials

For the quantification of the metal content, the fiber materials were cut 2 × 2 cm^2^ and incubated in 1 mL of HNO_3_ (Nitric Acid TraceMetal Grade 67–69%, Seastar Chemicals) for 24 h at 65 °C. After incubation, the suspension was vortexed and diluted 1:1000 in 1% HNO_3_. The concentrations of Cu and Ag in diluted suspensions were analyzed by Atomic absorption spectroscopy (AAS) (contrAA 800, Analytik Jena Ag, Analytik Jena GmbH, Jena, Germany).

For the quantification of released metal content, the 2 × 2 cm^2^ of filter material was added into 3 mL of DI water for 5 or 20 min and incubated at 36 °C on a shaker. After incubation, the supernatant was dissolved (1:9) in HNO_3_ (Nitric Acid TraceMetal Grade 67–69%, Seastar Chemicals) for 10 min, at room temperature, and vortexed, followed by the dilution of the solutions 100 times in 1% HNO_3_ and analysis by AAS (contrAA 800, Analytik Jena Ag).

### 2.5. Cytotoxicity Assay

The cytotoxic effects of CuO, ZnO and Ag NPs and corresponding salts were studied by using the WST-1 assay. WST-1 reagent (Roche) was used according to the manufacturers’ protocol. The cells (MDCK-2, Vero-E6, or ST) were grown to approximately 70% confluency on 96-well plates. The growth medium was removed, and metal NPs or salts at specified concentrations were diluted in the growth medium at 100 µL/well. Then, 48 h later, WST-1 reagent was added (10 µL/well), the cells were incubated for 3 h, and the optical density (OD) at 450 nm was determined by a Tecan Sunrise spectrophotometer. Data analysis was carried out by averaging the OD reads of triplicate wells, subtracting blank (wells without cells), and comparing the derived values with the 0-controls (cells treated with growth medium without NPs). The controls’ viability was considered to be 100%, and cell viability <80% was considered to be an indication of the cytotoxicity of the used materials.

### 2.6. Antibacterial Assay

*Escherichia coli* MG1655 strain was obtained from the genetic stock center (Yale University), and *Staphylococcus aureus* 6538 strain was obtained from the ATCC. Before the tests, the bacteria were cultivated in 3 mL of RPMI, at 37 °C, with shaking at 200 rpm, overnight. Then 400 µL of overnight cultures was mixed with 20 mL of LB broth and incubated for 4 h to reach the exponential growth phase. After the incubation, the OD at 620 nm (OD620) was measured, and bacterial suspension at absorbance 0.1 (measured at OD 620) was prepared. A total of 500 µL of suspension was spread on the square Petri dish (10 cm × 10 cm) with LB agar. On the freshly inoculated LB agar, the filter materials were placed, followed by incubation for 24 h at 37 °C; the antibacterial effect of the materials was then evaluated by visual inspection.

### 2.7. Assessment of Antiviral Activity of Nanomaterials and Corresponding Salts in Suspensions

Virus stocks in cell culture medium were diluted 10 times in water; virus titers were 6.65 × 10^4^ pfu/mL for influenza A virus, 8.35 × 10^6^ pfu/mL for TGEV, and 2.5 × 10^5^ pfu/mL for SARS-CoV-2. The CuO-COOH, CuO-NH_2_, CuO, ZnO, and Ag nanomaterial suspensions and corresponding salts’ (CuSO_4_, AgNO_3_, and ZnSO_4_) dilutions were prepared in distilled water. Viruses and nanomaterials at specified concentrations were mixed 1:1 and incubated for 1 h, at room temperature. After incubation, the viruses were titrated by plaque assays. For this, the 10-fold dilution series in VGM was prepared from the virus–nanomaterial mixtures, starting from 10^−1^ to 10^−4^ dilutions.

In the case of the influenza A virus and TGEV, 100% confluent cells on 12-well plates were washed with phosphate buffered saline (PBS) and infected with 150 µL dilution per well in duplicate for 1 h at 37 °C and 5% CO_2_ in a humidified atmosphere, gently rocking the plates every 15 min. Then infection inoculum was removed and replaced with VGM containing 1% carboxymethylcellulose (CMC); in the case of the influenza A virus, the mixture was supplemented with TPCK-trypsin at a final concentration of 1 µg/mL. At 96 h post-infection, the plates were stained with crystal violet, washed, and dried; the plaques were then visually counted, and the virus titers were calculated.

For SARS-CoV-2, 100% confluent Vero E6 cells grown on the 96-well plates were infected with 25 µL dilution per well in duplicate for 1 h at 37 °C and 5% CO_2_, and then they were layered with 1% CMC in VGM and incubated for 48 h at 37 °C and 5% CO_2_ in a humidified atmosphere_._ The plates were fixed with ice-cold 80% acetone in PBS for 1 h at −20 °C and air-dried for at least 1 h. The plates were blocked with the Thermo Scientific blocking buffer (Ref. No. 37587) and then stained with rabbit monoclonal anti-SARS-CoV-2 nucleocapsid antibody (Icosagen Ltd., R1-166-100, Tartu, Estonia) and secondary anti-rabbit IRDye800-conjugated antibody (LI-COR). The stained plates were read at 800 nm, using LI-COR Odyssey machine; the immuno-stained plaques were counted in the wells with visually distinguished single plaques, and the virus titers were calculated by taking into account the volume of viral inoculum. The data were further analyzed with GraphPad Prism 9.0 software (San Diego, CA, USA) to calculate the half maximal inhibitory concentration (IC50) for each virus and nanomaterial combination.

### 2.8. Deactivation of Viruses by Fiber Materials

The electrospun fiber materials were cut into 2 × 2 cm pieces and weighed to ensure that the weight difference of pieces was less than 10%. The autoclaved material pieces were placed onto 12-well plates. Then 30 µL of TGEV (2.5 × 10^5^ pfu-s) or influenza A virus (1.8 × 10^4^ pfu-s) was pipetted onto material, and 470 µL DMEM (without additives) was added. The sample was collected immediately (5 min time point), or the material piece was incubated with virus for 1 h at 28 °C, 5% CO_2_, in a humidified atmosphere. The samples were collected from the well by pipet. Then a material piece was placed in a 1.5 mL tube; centrifuged for 3 min at 13,400× *g*; and the liquid samples were combined, mixed, and titrated by plaque assay.

To evaluate the antiviral activity of materials against SARS-CoV-2, the materials were cut into 2 × 2 cm pieces and placed into 50 mL falcon tubes. Then 30 µL of SARS-CoV-2 virus stock was added. For the 0 h time point, the sample was processed immediately, and for the 1 h time point, the samples were incubated at 28 °C by adding 470 µL of PBS, followed by brief vortexing and centrifugation at 3200× *g* for 3 min at room temperature. Supernatants were collected to the tubes, and 10-fold dilutions were prepared in VGM (final volume of 200 µL). For the plaque assay, the Vero E6 cells seeded on a 96-well plate on the previous day were incubated with 100 µL diluted sample/well for 1 h at 37 °C in a humidified 5% CO_2_ atmosphere. After incubation, the cells were overlaid with VGM/1% CMC solution and incubated for 48 h at 37 °C in a humidified 5% CO_2_ atmosphere. The cells were fixed with ice-cold 80% acetone/PBS at −20 °C for 1 h, followed by an immunoplaque assay with anti-SARS-CoV2-N 82C3 (Icosagen Ltd., R1-166-100), as described above.

### 2.9. Statistical Analysis

IC50 was calculated by using GraphPad Prism 9.0 software, and *p*-values were calculated in Excel, using Student’s *t*-test.

## 3. Results

### 3.1. Antiviral Efficacy of Nanoparticles in Suspensions

The physicochemical properties of NPs were characterized by our group before [41,42] and are summarized in Appendix A. The antiviral properties of metal NPs and corresponding salts were investigated by a plaque assay first in suspensions against influenza A virus and coronaviruses TGEV and SARS-CoV-2. TGEV was used as a model for coronavirus susceptibility evaluation—as a safer Biosafety Level 2 (BSL2) model for a coronavirus. Metal salts were used as controls to the NPs tested. The absence of cytotoxicity of tested compounds was confirmed on cell lines in vitro since the infection of cells by virus in suspension with tested compounds was an integral part of the plaque assay. The testing of metal compounds’ antiviral efficacies in suspensions was performed at the 1-hour time point, as we were aiming to generate and test materials with high efficacies.

The testing in suspensions showed that the most potent metal salt against all viruses, i.e., A/WSN/1933 (H1N1), SARS-CoV-2, and TGEV, was CuSO_4_ (IC50 1.40, 0.45, and 4.44 mg/L, respectively) (Table 1), indicating the superiority of copper and, more precisely, copper ions in the antiviral testing setup. Please refer to the dose response curves in Appendix A. Thus, CuO, CuO-NH_2_, or CuO-COOH were less effective against tested viruses than CuSO_4_ (with the exception of CuO-COOH that has IC50 value 0.57 mg/L for A/WSN/1933 (H1N1)) (Table 1). This might be due to the fact that carboxylation has additional antiviral efficacy, as COOH groups readily bind to viral RNA [43].

Silver is a well-known antibacterial and a promising agent in the fight against coronavirus, whereas, in our study, the AgNO_3_ salt was not effective even in the highest concentration tested (100 mg/L) and did not reduce the titers of A/WSN/1933 (H1N1) and TGEV enough to calculate IC50. The other metal in the study, Zn salt, ZnSO_4_, was less effective against the tested influenza and coronavirus compared to CuSO_4_ (Table 1). Hence, we selected copper compounds as a salt CuSO_4_ and CuO NPs for incorporation into filter materials.

### 3.2. Development and Characterization of Filter Materials

#### 3.2.1. Compatibility of NPs with Solvent Systems for Preparation of the Filter Materials

Next, CuSO_4_ and CuO NPs were incorporated into 17% CA polymer to produce filter materials. First, the compatibility of CA polymers in dispersions of acetone–DMAc (Ac-DMAc) solvent systems with NPs and metals was tested. The NH_2_ and COOH functionalized and unfunctionalized CuO NPs in suspensions and CuSO_4_ were evaluated by DLS in solvent systems compatible with CA for 24-h and 1-week time points (Appendix A). The functionalized NPs CuO-NH_2_ and CuO-COOH formed aggregates in solvents, as exemplified by the high polydispersity index (PDI) values (0.7–1) and, thus, were not suitable for electrospinning. Hence, surface-functionalized NPs were not used to produce fiber materials, whereas the unfunctionalized CuO NPs, and especially CuSO_4_ salt, demonstrated excellent compatibility to be incorporated into CA filter materials and were used for generating the fiber materials.

#### 3.2.2. Development of Filter Materials

Filter materials were produced by using electrospinning techniques and CA as a polymer carrier for CuSO_4_ and CuO NPs (Table 2). Moreover, thymol was used for the electrospinning of filter materials to raise the porosity and hydrophilicity of the nanofibers, as previously described [44]. We hypothesize that by increasing porosity, thymol will improve metal release and, thus, antimicrobial properties of filter materials, and by raising the hydrophilicity, we will obtain better release of the metal ions to inactivate the virus efficiently. The developed filter materials were named as follows: CA (CA without antimicrobial components); CA_7.5%CuSO_4_ (CA with 7.5% CuSO_4_); CA_10%CuO (CA with 10% CuO); CA_thymol (CA with 10% thymol); and CA_thymol_7.5%CuSO_4_ (CA with 10% thymol and 7.5% CuSO_4_). All shown percentages were calculated based on Cu content.

The maximum nominal concentration of Cu from CuSO_4_ and CuO that we were able to incorporate into CA was up to 8% (Table 3). Namely, when 7.5% of Cu from CuSO_4_ was added to CA polymer solutions (sample designated CA_7.5%CuSO_4_), the measured concentration of Cu in the final filter material was 4.68%. Thymol remarkably improved the incorporation of CuSO_4_ into CA; with the addition of 7.5% of Cu from CuSO_4_ to CA with thymol, the measured concentration of Cu in this filter material was very close to nominal, 7.38%. The filter materials were characterized for fiber morphologies, material thickness, hydrophilicity/hydrophobicity, and air-filtration parameters (Table 2). Thymol improved the hydrophilicity of the materials and the release of Cu 1.7 times compared to CA fibers without thymol (Table 2 and Table 3).

#### 3.2.3. Morphology of Filter Materials

Scanning electron microscopy (SEM) was used to assess the morphology of fibers of filter materials (Figure 1 and Figure 2 and Appendix A). According to SEM, all fibers had random orientation (Figure 1 and Figure 2). The average diameter of the fibers was 600 nm, giving one of the biggest advantages of nanoscale: through the addition of a small quantity of metals or NPs to maximize specific functionalities on a high scale. We confirmed by EDX that filter materials with CuSO_4_ and CuO contained a high concentration of Cu (Figure 1D,F), whereas CA alone contained only traces of Cu (Figure 1B). In addition, CuO NPs were clearly visible in the case of CA with CuO NPs (Figure 1E). The addition of thymol in the CA materials did not increase the porosity of material, but it altered the surface structure of the fibers, giving more specific surface area for the material and virus interaction to take place (Figure 2).

#### 3.2.4. Thickness, Hydrophilicity, and Air Filtration Parameters of Filter Materials

The filtration performances were measured by determining the penetration of 1% NaCl aerosol particles, using an Aerosol generator TSI model 3076. The air permeability and the particle filtration efficiency were associated with the mat thickness having higher air permeability and filtration efficiency for mats with ca. 0.05 mm thickness. The thickness of the produced mats was very similar, being in the range of 0.04–0.06 mm with the exception of CA_10%CuO (0.163 mm). The small variation of this parameter can be explained with non-uniform distribution of additives in the electrospun material as the handmade electrospun device with rotating drum collector was used. In this study, the aim was to use electrospinning technique for producing protective material with nanosize fibers comprising different additives. The choice of the electrospinning parameters depended on the solution properties, viscosity, and the distribution/mixing/compatibility of the used additives in the polymer solutions. Thus, the changing of the flow rate enabled us to adjust to these parameters (Appendix A). Compared to the materials of CA without Cu compounds, the air permeability of Cu-containing CA materials decreased around 2.5 times but was still acceptable [45]. Thus, the materials had acceptable performance parameters and were used in further tests (Table 2).

### 3.3. Antibacterial Efficacy of Filter Materials Comprising Copper NPs and Copper Salt

In the next step, the antibacterial activity of developed filter materials was studied against *E. coli* and *S. aureus*. The antibacterial activity of the materials against bacteria was assessed on agar diffusion assay and is visualized in Figure 3. As expected, CA alone (control) did not inhibit the growth of bacteria. All other filter materials had excellent antibacterial activity against both *E. coli* and *S. aureus*. Interestingly, CA with thymol had an antibacterial effect even without the incorporation of Cu compounds. The best antibacterial effect (the largest bacteria-free zone) was observed in the case of CA with CuSO_4_ and CA with thymol and CuSO_4_.

### 3.4. Antiviral Efficacy of Fiber Materials Comprising Antimicrobial Metals

Previously, the antiviral efficacy was measured for nanoparticles in suspensions (Table 1). The antiviral activity of the produced filter materials against A/WSN/1933 (H1N1), SARS-CoV-2, and TGEV viruses was studied by incubating virus stock with the material and determination of the effect on virus titer by plaque assay. The best virucidal effect (virucidal effect is expressed as reduction of viral titers after 5 min and 1 h of exposure) was observed in the case of CA with CuSO_4_ and CA with thymol and CuSO_4_ (Figure 4). Materials comprising CuSO_4_ reduced influenza A virus titers 1.1–1.8 log already after 5 min of exposure and 1.6–1.8 log after 1 h of exposure (Figure 4A). From three tested viruses, the virucidal effect of all filter materials on TGEV was the lowest: 5-minute exposure resulted in no significant virus titer reduction, and mild 0.2 log reduction was observed after 1-hour exposure for CA material with CuSO_4_; and no effect was observed when thymol was the additive for the same material (Figure 4C). From the tested viruses, the best virucidal effect of filter materials was observed for SARS-CoV-2: CA fibers with 7.5% Cu from CuSO_4_ completely eliminated infectivity of SARS-CoV-2 after 1 h of contact exposure, and CA_thymol_7.5%CuSO_4_ material reduced the titer by 1.14 logs (Figure 4B). Interestingly, different from filter materials comprising CuSO_4_, the materials of CA with 10% CuO were not effective against any of the tested viruses (Figure 4). The addition of thymol alone to the CA material, however, reduced A/WSN/1933 (H1N1) titer approximately 0.7 logs compared to the CA control material without additives, but the difference was not statistically significant (Figure 4A). Thymol’s antiviral effect was limited to the influenza virus in our experiments, as the CA_thymol material did not reduce the titers of coronaviruses, indicating the importance of hydrophobicity in the antiviral efficacy [46]. We suggest that the influenza virus is more susceptible to released Cu ions, and SARS-CoV-2 is more susceptible to the hydrophobic interactions of the material. To summarize, similar to the antibacterial properties, filter materials with CuSO_4_ demonstrated the best antiviral properties (Figure 4), and, additionally, material hydrophilicity and hydrophobicity played a role.

## 4. Discussion

Influenza A virus is a respiratory virus that has caused several historic outbreaks, such as the Spanish flu in 1918, the 1977 Russian flu pandemic, and the 2009 swine flu pandemic. SARS-CoV-2 has caused over 600 million human infections and at least 6.5 million deaths in less than 3 years. The rapid spread of these viruses serves as evidence of the highly effective spread that occurs mostly (or at least partly) by the droplet-transmission ways [47,48]. Hence, the development of antiviral filter materials and face masks could be an option to suspend the transmission of these viruses, their variants, and other airborne pathogens. The use of antiviral and antimicrobial masks from environmentally friendly materials will significantly reduce the adverse environmental impact and the risk of contamination during their handling and disposal, as this is an important concern during the pandemic [49]. These masks could aid to protect the public health in community settings and in various healthcare settings. For example, the face masks based on these filter materials may be used in hospitals even beyond the pandemic caused by SARS-CoV-2 in order to prevent hospital-acquired infections. The bacteria *Escherichia coli* and *Staphylococcus aureus* are the major causes of hospital-acquired infections [50]. The pathogenic strains of *E. coli* have been associated with pneumonia and the pathogenic strains of *S. aureus* with pneumonia and meningitis. Due to the differential membrane structures of *E. coli* and *S. aureus*, they represent the Gram-negative and the Gram-positive bacteria, respectively. Thus, we used these two bacteria as models to study the antibacterial properties of the developed filter materials. To investigate the antiviral properties of the developed materials, we used two respiratory viruses: influenza A/WSN/1933 (H1N1) virus and SARS-CoV-2 and TGEV, a coronavirus mostly transmitted by oral route.

It has been reported that, depending on the surfaces, the human coronaviruses can remain infectious for up to 9 days [51], and influenza A virus can for up to 2 days [52], whereas survival is longest on the most common surfaces, such as wooden, stainless steel, and plastic. One way is to tackle the transmission of pathogens by impairing the infectious properties of these viruses and bacteria by new antimicrobial materials. The virucidal properties of fiber materials comprising metals have been investigated in previous studies. For example, Hodek et al. showed that protective hybrid coating prepared via sol–gel method and applied on glass slides or into the wells of polymethyl methacrylate plates containing silver, copper, and zinc cations effectively inactivated HIV and other enveloped viruses [33,53]. That brought us to the current study to test Ag, Zn, and Cu compounds as potential antivirals against SARS-CoV-2, TGEV, and other enveloped viruses first in suspensions and to use the most promising of them to incorporate into polymers to produce filter materials for face masks. These filter materials were intended to inactivate respiratory viruses, as well as common bacterial pathogens causing hospital acquired infections.

In our study, the metals were first tested in the suspensions. Somewhat surprisingly, Ag, the most widely used antibacterial metal, was not efficient to inhibit viruses used in our tests that showed (in suspensions) IC50 values >100 mg/L for AgNO_3_ against A/WSN/1933 (H1N1) and TGEV, and >1000 mg/L for Ag NP-s against influenza A virus. The reason could be that the antibacterial activity of Ag und Cu is strongly attributed to the release of ions and their effect on cellular components and processes. In this regard, virus particles having no metabolic activities are drastically different from bacteria. Ag ions interfere with metabolic processes such as energy generation and cell proliferation, which are important for bacteria but not for virus particles. In contrast, Cu ions have been reported to damage the virus capsid and envelope [54]. Indeed, Cu compounds were effective against influenza A virus from 0.57 to 49.25 mg/L, depending on the compound and functionalization of the NPs, and they showed the lowest IC50 for CuO-COOH (Table 1). Of the tested Cu compounds, CuSO_4_ was the most efficient against SARS-CoV-2 (IC50 0.45 mg/L) and, to a lesser extent, against TGEV (IC50 4.44 mg/L). In our experiments influenza virus was more sensitive to CuO NPs with or without NH_2_ or COOH modifications than coronaviruses. The high sensitivity of this virus to these compounds is most probably due to the rapid release of Cu ions from CuO-COOH [41] and COOH properties to effectively bind RNA of SARS-CoV-2 [55], thus impairing the propagation of the virus. The CuO-NH_2_ actively releases the Cu ions and generates more ROS damaging the virus compared to unfunctionalized CuO [41]. Zn compounds were less effective than Cu compounds against A/WSN/1933 (H1N1), having IC50 values from 3.39 to 134.8 mg/L after 1 h of contact. Zn is also generally less potent metal against bacteria as compared to Cu and Ag [19,56], and its modest antiviral activity in the current study is in concurrence with the previous report where ZnO had 50% activity against influenza A virus at 75 mg/L concentration after 4 h of exposure [57].

Our findings confirmed the excellent antiviral properties of copper and are in agreement with those of multiple studies [23,34,36,58,59,60,61,62,63]. This might be due to copper’s capability to generate reactive oxygen species (ROS) in a moist environment, which oxidizes the capsid proteins of viruses [64]. Cu destroys the replication abilities by interacting with the hereditary material and propagation abilities of SARS-CoV and disturbs the host and virus interaction and, thus, internalization of the viral particles of influenza and other respiratory viruses [65]. Fujimori et al. investigated the antiviral activity of copper iodide (CuI) nanoparticles of 160 nm in size against swine influenza virus, whereas CuI nanoparticles inactivated the virus in a dose-dependent manner, and the total inactivation was observed after 1 h of contact for 100 mg/L concentration [23]. The antiviral activity of CuI nanoparticles was mainly attributed to the release of one-valent copper ions, which lead to the generation of ROS that caused the oxidation of viral proteins and the damage of the viral RNA. Thus, there is a high potential of copper to be used as an antiviral agent in face masks to impair the airborne disease transmission of influenza and coronaviruses in hospitals and in household settings [62].

For enhanced effects and targeted applications, CuSO_4_ and CuO NPs were incorporated into polymers to develop functional antimicrobial materials. Compared to traditional biocides, antimicrobial polymers have advantages such as enhanced antimicrobial activity, prolonged release of active substances, and the avoidance of resistance [66,67]. Based on the literature data and on our tests of different metals and NPs in suspensions (Table 1), we selected Cu compounds for incorporation into CA, a modified natural polymer that is biodegradable and therefore has environmentally friendly properties. As a fiber, it is used in textiles for its low price, softness, comfort, ease for modifications, and ability to provide controlled release of incorporated compounds [68,69]. Therefore, we developed CA filter materials comprising CuO and CuSO_4_ and tested them for antibacterial and antiviral activities. CuO was chosen as an additive to the polymer because of the tested Cu compounds the COOH- and NH_2_-functionalized CuO NPs had excellent antiviral properties in suspension. To produce face masks with copper, COOH- NH_2_-functionalized and unfunctionalized CuO NPs and CuSO_4_ were first tested for their compatibility with CA polymer and solvent systems. Acetone and DMAc were selected as solvents, since DMAc is readily adjustable to electrospinning with a wide range of polymers for filtering purposes and acetone is added to the solvent system to facilitate fiber formation and filter material synthesis in electrospinning. However, both CuO NPs with surface functionalization demonstrated incompatibility with the selected solvent system by forming large aggregates, as reflected in large hydrodynamic size and high polydispersity indexes (Appendix A).

Hence, CA filter materials with CuSO_4_ and CuO NPs were produced, with or without thymol demonstrating good air permeability and particle filtration properties (Table 2). All materials with Cu compounds posed high antibacterial activity against *E. coli* and *S. aureus*, with the CuSO_4_-based material having the highest efficacy (Figure 4). While CuO NP-based filter materials did not exhibit antiviral properties, CuSO_4_-based filter materials showed some efficacy against A/WSN/1933 (H1N1) (titer reduced ca. 1.7 log) and differing efficacy against coronaviruses after 1 h of contact. The high efficacy of CuSO_4_-containing materials against bacteria and viruses was due to the high release of Cu from the materials containing CuSO_4_ (46–78% after 1 h), whereas no release of the ions was detected for fibers containing CuO (Table 3). Since tests with SARS-CoV-2 were very demanding and necessitated BSL-3 conditions and infrastructure, we also attempted to use coronavirus TGEV as a model coronavirus to screen compounds and filter materials before tests with SARS-CoV-2. Antiviral properties of CuSO_4_-based filter materials against TGEV were low: less than 1 log reduction in titer after 1-h exposure. In contrast, the SARS-CoV-2 titer decreased to the point of being not detectable after exposure to CA_7.5%CuSO_4_ material and approximately 1.1 log after 1 h of exposure to CA_thymol_7.5%CuSO_4_ material (Figure 4). Our data indicate that, as a surrogate model for SARS-CoV-2, TGEV is inferior to no-related influenza A virus that uses the same entry route as SARS-CoV-2. We assume the difference in SARS-CoV-2 and TGEV susceptibility to CuSO_4_ containing materials is because TGEV similarly to other viruses that spread in fecal–oral way is in general less susceptible to antivirals, since it is adapted to the harsh conditions of gastrointestinal tract [70].

To improve the antiviral efficacy of the material, thymol was added to CA. Thymol is considered to be an excellent additive as it possesses inherent antibacterial activity as shown by others [71] and in our study (Figure 3). Moreover, thymol facilitates material hydrophilicity that in general enhances antiviral activity [72,73,74,75] and porosity that increases the active surface area [44]. Thus, we used the thymol to increase hydrophilicity, hypothesizing that it will increase the release of antiviral Cu ions.

Indeed, in our study addition of thymol increased the hydrophilicity of filter materials (Table 2) but did not improve their antiviral and antibacterial properties (Figure 3 and Figure 4). Surprisingly, in case of SARS-CoV-2, thymol even reduced the antiviral properties of CA with CuSO_4_ 1.14 log after 1 h of contact (Figure 4B). From the literature, it is known that hydrophobic properties of materials enable a better contact with the lipid parts of viruses and bacteria, whereas the hydrophilic properties facilitate a better release of metal ions from the fibers [72,75]. Indeed, in our study, the addition of thymol increased the release of Cu ions (Table 3) but did not improve antiviral properties. Based on these data, we assume that, in addition to Cu release from filter materials, the major factor determining antiviral properties is hydrophobicity. We suggest that hydrophobicity of the material determined antiviral efficacy against SARS-CoV-2 (hydrophobic material without thymol was more effective) and is consistent with the previous literature [18]. Presumably, the effect of hydrophobicity/hydrophilicity is different for different viruses, depending on the properties of their envelope [33,53]. SARS-CoV-2 has a hydrophobic lipid envelope, and we assume that hydrophobic filter materials enabled better virus–material contact, enhancing antiviral properties of material.

The successful antiviral efficacy of materials comprising copper compounds that was demonstrated in this study is consistent with the previous literature. For example, masks containing ca. 2% weight/weight (*w*/*w*) copper particles spunbond in polypropylene fabric in fiber layers showed antiviral properties: a 2.88 log and 3.13 log additional reduction of infectivity of human influenza A virus were observed after 30 min of exposure as compared to the control masks without copper nanoparticles [58]. To the best of our knowledge, biopolymer-based CuSO_4_-containing CA filter materials developed in this study are unique and have not been reported in the literature. The CA filter material containing 7.5% CuSO_4_ had excellent antibacterial properties (Figure 3), completely inactivated SARS-CoV-2 (Figure 4B), and caused 1.6 log reduction of A/WSN/1933 (H1N1) virus titers (Figure 4A) after 1 h of exposure, providing broad-spectrum protection from air-borne pathogens. These properties are similar and consistent with previously described analogous materials. Self-sterilizing photoactive nanomask composed of biopolymer Shellac showed ca. 2–3 log decreased viability of *E. coli* after 1 h of exposure and loss of infectivity of virus-like particles after 5 min of exposure [76]. The mask fibers containing licorice root in the composition of biocompatible polymers [77] and face masks containing copper [78] or quaternary ions [79] have also been demonstrated to possess antiviral efficacies against SARS-CoV-2 virus. In the study of Jung et al., the authors coated the filter layer of commercially available face masks from synthetic polypropylene with a thin film of copper and demonstrated that SARS-CoV-2 titer was reduced by more than 75% after 1 h of exposure to this mask [78].

In summary, the incorporation of CuSO_4_ into filter materials is a promising mean to obtain materials that prevent spread of enveloped viruses, such as influenza viruses; coronaviruses, including SARS-CoV-2; and bacteria such as *E. coli* and *S. aureus*.

## 5. Conclusions

Here we presented valuable data on virucidal effects of Zn, Cu, and Ag salts and respective NPs to three viruses, TGEV, influenza A virus, and SARS-CoV-2. Out of the tested compounds, the CuSO_4_ and CuO NPs had the best antiviral properties and compatibility with the electrospinning technique, followed by Zn and Ag, and thus, they were selected to develop novel filter materials as an intervention to mitigate the spread of pathogens such as viruses A/WSN/1933 (H1N1), SARS-CoV-2, and TGEV and the bacteria *E. coli* and *S. aureus*. As a main result of the study, novel filter materials based on biopolymer cellulose acetate and incorporating CuSO_4_ or CuO NPs were developed. CuSO_4_-based filter materials were the most effective against SARS-CoV-2 and reduced virus titers already after 5 min of exposure and completely inactivated the virus after 1 h of exposure. Our data on antiviral and antibacterial properties of metals, metal-based NPs, and filter material thereof will help to govern the future development of advanced antibacterial and antiviral materials and surfaces.

## Figures and Tables

**Figure 1 pharmaceutics-14-02549-f001:**
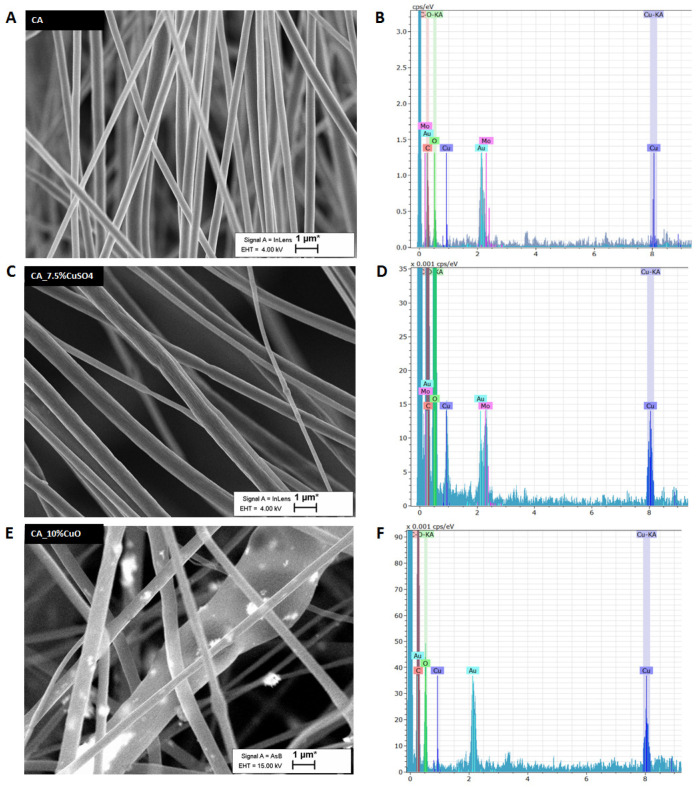
Morphological characteristics by SEM imaging for electrospun CA filter mats (panels **A**,**C**,**E**) and EDX analysis of the mats (panels **B**,**D**,**F**). Please refer to the different scales on the *y*-axis.

**Figure 2 pharmaceutics-14-02549-f002:**
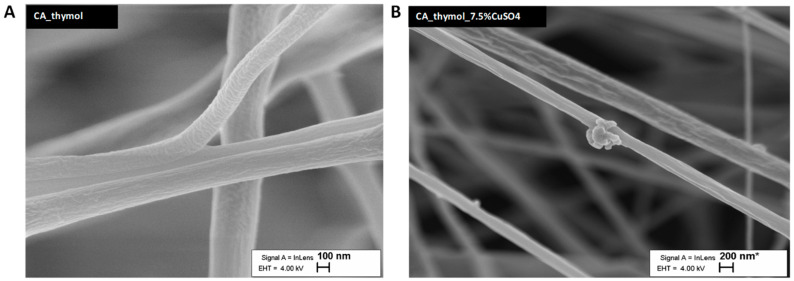
Morphological characteristics by SEM imaging for electrospun CA filter mats comprising thymol (**A**) and CuSO_4_ (**B**) as additives, showing the texture change on the surface.

**Figure 3 pharmaceutics-14-02549-f003:**
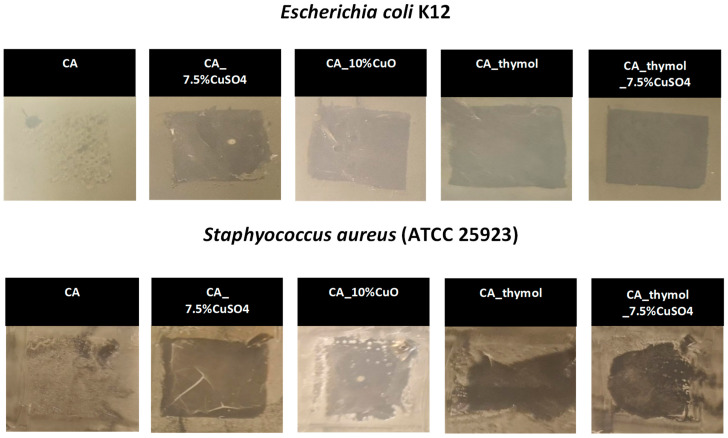
Antibacterial activity of filter materials against bacteria *Escherichia coli* and *Staphylococcus aureus.* The filter materials were removed for the images to demonstrate the bacteria-free area under the materials.

**Figure 4 pharmaceutics-14-02549-f004:**
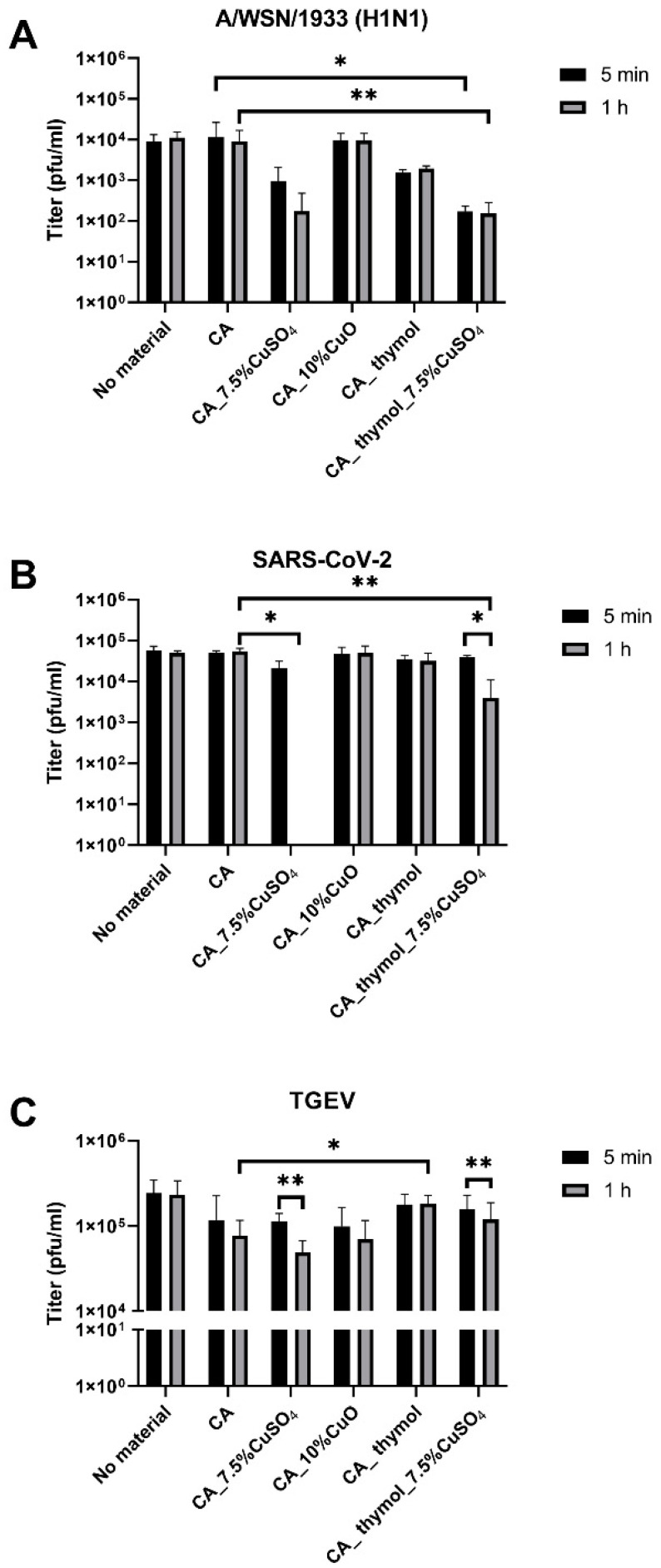
Antiviral properties of CA without additives (control), 7.5% CuSO_4_ and 10% CuO in CA against influenza A/WSN/1933 (H1N1) virus (**A**), SARS-CoV-2 (**B**), and transmissible gastroenteritis coronavirus TGEV (**C**). Statistical significance is represented as follows: * *p* < 0.05; ** *p* < 0.01.

**Table 1 pharmaceutics-14-02549-t001:** Antiviral efficacy of metal salts and metal nanoparticles against influenza A virus, SARS-CoV-2, and TGEV in water suspensions.

Substance	IC50 (mg/L)		
	Influenza A Virus	SARS-CoV-2	TGEV
CuSO_4_	1.40	0.45	4.44
CuO	49.25	>100	383.4
CuO-NH_2_	1.88	149.1	8.8
CuO-COOH	0.57	79.68	13.75
ZnSO_4_	3.39	35.65	ND
ZnO	134.8	ND	ND
AgNO_3_	>100	NA	>100
Ag NP	>1000	NA	NA

NA—not applicable: since Ag NPs and AgNO_3_ were toxic to VeroE6 cells used for tests with SARS-CoV-2 already at 10 mg/L. ND—not determined.

**Table 2 pharmaceutics-14-02549-t002:** Characteristics of electrospun filter materials.

Samples	Mat Thickness	SEM Diameter	Air Permeability	Aerosol Filtration Efficiency, %	Aerosol Filtration Efficiency, %	Hydrophobic/ Hydrophilic Measuring	Contact Angle Measuring
	mm	nm	Pa/cm^2^	whole range 11.8–429.4 nm	300 nm		°
CA	0.051	750	125.0	99.3	99.6	hydrophobic	104
CA_7.5% CuSO_4_	0.062	972	54.1	84.3	85.5	hydrophobic	107
CA_10% CuO	0.163	759	47.4	78.4	81.6	hydrophobic	99
CA_thymol	0.036	545	45.4	ND *	ND	hydrophilic	82
CA_thymol_7.5% CuSO_4_	ND	431	55.9	ND	ND	hydrophilic	86

* ND—not determined, CA—cellulose acetate.

**Table 3 pharmaceutics-14-02549-t003:** The metal content and release of copper ions from CA filter materials.

Fiber Material	Cu Content, %	Released Cu Content, 1 h, %
CA	0	ND
CA_7.5%CuSO_4_	4.68 ± 1.1	46
CA_10%CuO	8.01 ± 0.39	0
CA_thymol	0	ND
CA_thymol_7.5%CuSO_4_	7.38 ± 0.61	78

ND—not determined.

## Data Availability

Not applicable.

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
