# Peer review of "Antibacterial and Antiviral Effects of Ag, Cu and Zn Metals, Respective Nanoparticles and Filter Materials Thereof against Coronavirus SARS-CoV-2 and Influenza A Virus"

_pharmaceutics, 2022, doi:10.3390/pharmaceutics14122549_

Round 1

Reviewer 1 Report

This paper introduces the impact of different metallic nanoparticles hosted by electrospun cellulose acetate nanofibers on different bacteria types and viruses. The topic is very interesting and important. However, the authors did not pay enough attention to the quality of presentation. Therefore, the paper can not be published on the final form due to the following serious concerns:

1- The abstract has to be one summarized paragraph. It has to be revised. 2- The authors have to handle the manuscript with care. Different font types and sizes are found. Different typos should be revised such as thickness unit in Table 2. 3- The first sentence in the introduction is totally misleading and irrelevant. It has to be omitted. Also, the first paragraph should include more that one reference. 4- Why do the authors use some hydrophilic membranes, given that it may not be suitable as filters? Contact angle measurements have to be added in the table. 5- To compare the filtration efficiency, you should control the thickness to be quite unified or closer to each other. Try to unify the electrospinning parameters or discuss why some parameters are not unified such as the flow rate. 6- Move some of the electrospinning process parameters "in brief as 1-2 statements" from supplemental file to the main manuscript. Keep the table AS IS in the supplemental file. 7- You should add nanofibers size for each SEM. 8- The photos of agar should be zoomed out to see clearly the bacteria-free zones. 9- The curves of antiviral effect have to be moved from supplemental file to the manuscript to show the clear antiviral effect, as the bar diagram did not show a clear change compared to the curves. Also, why did not the authors try extended time for testing? 10-  What is the new contribution that the authors did compared to other similar trends in the literature? 11- The flow of paragraphs in discussion part is not correlated together.    

Author Response

Thank you for your valuable suggestions. The answers are below:

  • The abstract has to be one summarized paragraph. It has to be revised.

The abstract was summarized and revised.

  • The authors have to handle the manuscript with care. Different font types and sizes are found. Different typos should be revised such as thickness unit in Table 2.

We apologize for the confusion as it seems that the online version converted the file in different fonts. This has been taken care of.  We also did our best to revise all the typos.

  • The first sentence in the introduction is totally misleading and irrelevant. It has to be omitted. Also, the first paragraph should include more that one reference.

Thank you for addressing this issue. The first two sentences in the introduction have been removed and in the first paragraph more of the relevant references have been included.

  • Why do the authors use some hydrophilic membranes, given that it may not be suitable as filters? Contact angle measurements have to be added in the table.

The authors hypothesized that the more hydrophilic the materials are the better they are wetted and enable more efficient release of the metal ions. Please refer to the contact angle measurements presented in numbers in Table 2.

  • To compare the filtration efficiency, you should control the thickness to be quite unified or closer to each other. Try to unify the electrospinning parameters or discuss why some parameters are not unified such as the flow rate.

An explanation (below) was added to the results paragraph „Thickness, hydrophilicity and air filtration parameters of filter materials“

„The thickness of the produced mats was very similar, being in the range of 0.04-0.06 mm with the exception of CA_10%CuO (0.163 mm). The small variation of this parameter can be explained with nonuniform distribution of additives in the electrospun material as the handmade electrospun device with rotating drum collector was used. In this study, the aim was to use electrospinning technique for producing protective material with nanosize fibers comprising different additives. The choice of the electrospinning parameters depended on the solution properties, viscosity and the distribution/ mixing/ compatibility of the used additives in the polymeric solutions. Thus, the changing of the flow rate enabled us to adjust to these parameters.“

  • Move some of the electrospinning process parameters "in brief as 1-2 statements" from supplemental file to the main manuscript. Keep the table AS IS in the supplemental file.

Thank you for addressing the issue. The text is amended accordingly.

  • You should add nanofibers size for each SEM.

 The average sizes of SEM diameters have been added to Table 2.

  • The photos of agar should be zoomed out to see clearly the bacteria-free zones.

               The photos have been magnified to present the bacteria-free zones in more detail.

  • The curves of antiviral effect have to be moved from supplemental file to the manuscript to show the clear antiviral effect, as the bar diagram did not show a clear change compared to the curves. Also, why did not the authors try extended time for testing?

The interpretation of the results was misleading. The „Antiviral efficacy of fiber materials comprising antimicrobial metals“ was rewritten for clarification. Briefly, the curves of antiviral effect in supplementary materials Figures 1, 2, and 3 were for metal salts and metal nanoparticles testing in suspensions. Based on these data, Table 1 has been compiled. Thus, the figures and Table 1 would duplicate each other if were both in the main text.  If acceptable, we would prefer to keep the dose-response curves in supplementary material and condense the results section by presenting data in Table 1.

The times for exposure were chosen because the authors wanted to test the efficacy of materials for no longer than 1 hour period of exposure. Also, the aim of the study was to generate efficient antimicrobial materials with high efficacy during a short period of time. Thus, the times for material exposures were 5 minutes and 1 hour. We added explanation to the results section. Hopefully this explanation is sufficient to address this issue. 

  • What is the new contribution that the authors did compared to other similar trends in the literature?

The new contribution is that instead of testing only one metal /nanoparticle at a time, several metals and nanoparticles with different surface functionalizations were compared for their antiviral and antibacterial efficacies. Of the tested metals (Ag, Zn and Cu as salts and in the form of nanoparticles), copper had the best antiviral effect and was used to produce materials.

The flow of paragraphs in discussion part is not correlated together.   

We did our best to harmonize the paragraphs together in a better way. Please refer to the revised text (the modifications are highlighted on yellow background).

Reviewer 2 Report

The paper presents interesting findings and can be accepted after major revision. The following issues should be clarified.

The results presented in the paper should be compared with previous works in a more complex manner. Numerous interesting and important papers which are closely related to these studies are abroad of the work. At least two excellent reviews can be mentioned. 

https://doi.org/10.1016/j.cej.2022.137048

https://doi.org/10.34133/2020/7286735

Unfortunately but in numerous places the authors only constant the fact without explanation of the observed results. 

Why does CuSO4 have more antivirus efficiency than other types of cuprous-containing substances? Appropriate discussion should be proposed.

Why are the virucidal properties of the NH2 and COOH functionalized and unfunctionalized CuO NPs different? Appropriate discussion should be proposed.

Why thymol did not improve the antiviral and antibacterial properties of the filter materials but in the case of SARS-CoV-2, thymol even reduced the antiviral properties of CA with CuSO4. Appropriate discussion should be proposed.

It is unclear to me, why thymol was used for the fabrication of the nanofibers? Is it related to raising the porosity of the nanofibers? How does thymol improve porosity? Appropriate discussion should be proposed.

Author Response

Thank you for your valuable suggestions. The answers are below:

The paper presents interesting findings and can be accepted after major revision. The following issues should be clarified. The results presented in the paper should be compared with previous works in a more complex manner. Numerous interesting and important papers which are closely related to these studies are abroad of the work. At least two excellent reviews can be mentioned. 

https://doi.org/10.1016/j.cej.2022.137048

https://doi.org/10.34133/2020/7286735 

Thank you for the note and for suggesting these very interesting articles. These and other articles have now been included in the revised manuscript.

Unfortunately but in numerous places the authors only constant the fact without explanation of the observed results.     

We added the explanations (highlighted in the yellow background).

Why does CuSO4 have more antivirus efficiency than other types of cuprous-containing substances? Appropriate discussion should be proposed.

We assume this is because antivirus efficiency was mainly caused by dissolved ions and from all tested Cu-containing substances, CuSO4 was the most dissolving (Table 3). This discussion was added (paragraph 6).

„The high efficacy of CuSO4-containing materials against bacteria and viruses was due to the high release of Cu from the materials containing CuSO4 (46-78% after 1 h) whereas no release of the ions was detected for fibers containing CuO (Table 3).“

Why are the virucidal properties of the NH2 and COOH functionalized and unfunctionalized CuO NPs different? Appropriate discussion should be proposed.

Thank you, the discussion was added in paragraph 3.  

“The high sensitivity of this virus to these compounds is most probably due to the rapid release of Cu ions from CuO-COOH [41] and COOH properties to bind effectively RNA of SARS-CoV-2 [55], thus impairing the propagation of the virus. The CuO-NH2 actively releases antiviral Cu ions and generates more ROS damaging the virus compared to functionalized CuO  [41]”

Why thymol did not improve the antiviral and antibacterial properties of the filter materials but in the case of SARS-CoV-2, thymol even reduced the antiviral properties of CA with CuSO4. Appropriate discussion should be proposed.

Thank you, discussion was added. The discussion was also changed accordingly mostly by rewriting the first sentences to harmonize. Hopefully, these changes were sufficient to accomplish a fluent discussion.

It is unclear to me, why thymol was used for the fabrication of the nanofibers? Is it related to raising the porosity of the nanofibers? How does thymol improve porosity? Appropriate discussion should be proposed.

Thank you for this inquiry. The explanation is the following: the authors hypothesized that thymol was to a) change the surface of the fibers towards a more porous texture and b) change the material from hydrophobic towards hydrophilic. The addition of thymol was successful in changing the surface structure and the material properties towards more hydrophilic which in turn altered the antiviral properties of the fibers materials.

The relevant discussion was added as explained above.    

Round 2

Reviewer 1 Report

I would like to thank the authors for their intensive revisions. All of my comments have been well-addressed.

Reviewer 2 Report

The Authors have answered all of my issues, the quality of the paper was essentially improved and the paper can be accepted for publication in its present form.